# A Bibliometric Review of the Keap1/Nrf2 Pathway and its Related Antioxidant Compounds

**DOI:** 10.3390/antiox8090353

**Published:** 2019-09-01

**Authors:** Ana Paunkov, Dionysios V. Chartoumpekis, Panos G. Ziros, Gerasimos P. Sykiotis

**Affiliations:** 1Service of Endocrinology, Diabetology and Metabolism, Lausanne University Hospital and University of Lausanne, SA08/02/250, Ave de la Sallaz 8, CH-1011 Lausanne, Switzerland; 2Division of Endocrinology, Department of Internal Medicine, School of Medicine, University of Patras, 265 04 Patras, Greece

**Keywords:** Nrf2, Keap1, bibliography, sulforaphane, natural product, antioxidant, cytoprotection

## Abstract

Nrf2 is a master transcriptional regulator of antioxidant and cytoprotective pathways. Currently in its third decade, research on Nrf2 has expanded to encompass not only basic but also clinical studies. In the present bibliometric review, we employed the VOSviewer tool to describe the existing Nrf2 literature landscape. As of July 2019, 11,931 papers on Nrf2 were listed in the “Web of Science” database, with more than 1000 new papers published each year. As expected, terms related to oxidative stress and antioxidant molecules occur very often in the Nrf2 literature throughout the years. Interestingly, there is also a gradual increase in the occurrence of terms related to diseases or to natural compounds, the most prominent being sulforaphane, curcumin, and resveratrol that modulate the Nrf2 pathway. Going beyond molecular biology/biochemistry and related fields, Nrf2 research has begun to spread into more clinical areas like endocrinology/metabolism, cardiology, and nephrology, likely reflecting an increased interest in clinical applications of Nrf2 pathway activators. China has become the most prolific producer of Nrf2 papers the last five years followed by the USA and Japan, a reverse pattern compared to the past. In conclusion, Nrf2 is the subject of a globally active research field that keeps growing and extends from bench to bedside.

## 1. Introduction

The nuclear factor, erythroid 2-like transcription factor 2 (Nrf2) pathway orchestrates the expression of antioxidant and cytoprotective genes. Its activity is induced upon exposure to oxidative or electrophilic stresses, including the so-called indirect antioxidant compounds (e.g., sulforaphane) [1,2,3]. Targets of Nrf2 include glutathione synthesis/recycling genes (e.g., γ-glutamylcysteine synthetase currently known as glutamate-cysteine ligase) [4,5], genes involved in detoxication/conjugation reactions (e.g., glutathione S-transferase) [6], antioxidative genes (e.g., NAD(P)H quinone dehydrogenase 1) [7] and proteasome units [8]. Because oxidative stress is implicated in diverse cellular functions as well as in the pathogenesis of various diseases, Nrf2 has been increasingly attracting the attention of both basic and clinical researchers.

In the Nrf2 research community, it is well-known that historically one major research focus had been on basic studies trying to decipher the mechanism of Nrf2 regulation by its cytoplasmic inhibitor, kelch-like ECH-associated protein 1 (Keap1) [9]. Another major research focus has been to identify the molecular pathways in which it is involved as a transcription factor of major importance in cell homeostasis [10]. In parallel, the development of mouse models with loss [11] or gain of Nrf2 function [12] facilitated research on Nrf2 in various diseases at the preclinical level. Some examples in this ever-expanding area include protection against acetaminophen toxicity [13], carcinogenesis [14], obesity and diabetes [15], neurodegenerative diseases [16], and kidney disease [17], to mention but a few.

The discovery that natural compounds, such as sulforaphane from broccoli sprouts, can activate the Nrf2 pathway [18] has also instigated the initiation of clinical trials to test these substances in the form of a purified drug or as a dietary supplement [19,20]. To date, such clinical trials have focused on cancer chemoprevention [21,22], detoxication of environmental pollutants [23], metabolic disease [24,25] and relapsing forms of multiple sclerosis [26]. Other studies have investigated synthetic activators of Nrf2, for example in diabetic nephropathy [27].

During the last 10 years, several excellent reviews have been published about Nrf2 and have summarized the progress in Nrf2-related research through the perspective of basic [2,28,29,30,31,32,33] or translational-clinical research [1,16,19,34,35,36,37,38]. As we are now in the third decade of Nrf2 research, we sought to analyze and document in an objective manner how this field has been expanding. To this end, we evaluated the existing bibliography on Nrf2 by performing a bibliometric analysis focused on: (i) The most common terms mentioned in Nrf2-related papers, (ii) the natural compounds used as means to modulate the activity of the Nrf2 pathway, (iii) the research volume output by thematology and by country, including an assessment of impact.

## 2. Materials and Methods

### 2.1. Data Source and Search Strategy

The Web of Science online database (Clarivate Analytics, Philadelphia, PA) was accessed in July 2019 and the following search terms were employed in an “advanced search”: (ALL = (Nrf2 NOT “nuclear respiratory factor”)), “language: English” and “document type: “Article”. The database “Web of Science Core Collection” was selected. This eliminated papers related to nuclear respiratory factors 1 and 2 that are also commonly abbreviated as “Nrf1” and “Nrf2”, respectively, leading to confusion. Of note, inclusion or exclusion of additional search terms referring to Nrf2 synonyms such as NFE2L2, NFE2-L2, Nrf-2, etc.) had no impact on the search results. By selecting “custom year range” timespan was set to between 1990 and 2019, and this was subsequently sub-divided into four periods (1990–2005, 2006–2010, 2011–2015 and 2016–2019). Web of Science citation files for selected periods were downloaded as “full record and cited references” and saved in a “tab-delimited” file format. 

### 2.2. Data Analyses and Presentation

The citations extracted using the aforementioned search strategy were imported into VOSviewer (Centre for Science and Technology Studies, Leiden University, Leiden, The Netherlands) [39] for bibliometric analyses. The following options were selected during the import: “Create a map based on bibliographic data”, “read data from bibliographic database files”, “type of analysis: Co-occurrence”, “unit of analysis: All keywords” and “counting method: Full counting”. Using these settings, the software analyzes the words in the titles and abstracts of the selected publications, relates them to the number of documents in which they occur together, and visualizes them in the form of bubble maps. Each bubble represents a word or phrase. The top 5000 common words from the corpus of contemporary American English (link: https://www.wordfrequency.info/free.asp?s=y), were excluded from the analysis, as previously described [40]. The terms “Nrf2” and “Keap1” were also excluded from the visualizations, because they are directly referring to the original search term that we employed (i.e., Nrf2). Similar analyses were performed focusing on the countries that produce Nrf2-related research papers following the same method and selecting “citation” or “co-authorship” as “type of analysis”. To focus on natural products that may be related to Nrf2, all the terms from all subperiods from 1990 up to 2019 were checked manually to identify references to natural products. These terms were then imported into VOSviewer for visualization of the results, including calculation of the following parameters: “Averaged citations” or “averaged citation count” is the average number of citations received by the documents in which a keyword or a term occurs or which were published by a source (in the present case, by a country). “Averaged publication year” is the average publication year of the documents in which a keyword or a term occurs or the average publication year of the documents published by a source, an author, an organization, or a country. The “research areas” to which the papers belong were identified using “Web of Science” and specifically “Web of Science Categories”. Finally, charts illustrating the research areas and maps illustrating the countries from where the Nrf2-related papers derive were designed with Excel 2016 (Microsoft, Redmond, WA, USA).

## 3. Results

### 3.1. Output of Research on Nrf2 in the Form of Publications has been Increasing Continuously from 1994 to 2019

The aforementioned search for Nrf2 in “Web of Science” resulted in 11,931 publications. The first paper about Nrf2 that meets the search terms was published in 1994. At that time, it was not referring to its cytoprotective roles, but it was focused on a search for regulators of the β-globin gene [41]. Only after 2000 did the papers on Nrf2 reach at least a double-digit number per year. In 2005, the paper number reached 100. Since 2015, more than 1000 papers have been published per year (Figure 1A). It is thus evident that there is an ever-increasing activity in research in the Nrf2 field, as further indicated by the cumulative number of publications (Figure 1B).

### 3.2. Temporal Evolution of Bibliographic Terms Occurring in Nrf2-Related Publications

As the publications per year on Nrf2 are gradually increasing, the following shorter time periods were defined: First, from 1990 to 2005, because the first publication on Nrf2 appeared in 1994, and only few publications on Nrf2 came out over the first decade that followed. Then, five-year intervals were defined, namely 2006–2010, 2011–2015, and 2016–to date (July 2019). The term maps generated after analysis of each respective time period using VOSviewer are shown in Figure 2, Figure 3, Figure 4 and Figure 5. By observing these graphs, it is apparent that the main theme in Nrf2 papers over the years has been oxidative stress. This is evident from the relevant terms “oxidative stress”, “antioxidant response element”, “antioxidant”, “glutathione” and cytoprotective enzymes such as “glutathione s-transferase”, “γ-glutamylcysteine synthetase”, “NAD(P)H quinone dehydrogenase 1 (Nqo1)”.

The most common terms appearing in Nrf2-related publications reflect the models used for the study of the pathway. Terms related to mice and rats appear throughout the years (Appendix A), because these are the most frequently used models in Nrf2 studies, especially after the development of Nrf2 knockout mice and other related mouse models (mice with floxed alleles of Keap1, etc.). The organ that appears in the bubble maps throughout the years is the liver. Given the central role of Nrf2 in the regulation of antioxidant and cytoprotective pathways, and since the liver is the main detoxification organ of the body, a great many Nrf2 studies focus on liver toxicity and related gene expression, as reflected in Figure 2, Figure 3, Figure 4 and Figure 5 and Appendix A, where the liver is featured across time periods.

Studies on the role of Nrf2 in the pathophysiology or prevention of diseases started to appear mainly after 2006. One exception is the potential of Nrf2 as a target for chemoprevention, which has been recognized before 2005. Starting from 2006 to 2010, besides carcinogenesis, other diseases feature prominently in Nrf2-related studies, notably Parkinson’s disease, Alzheimer’s disease and chronic obstructive pulmonary disease (Figure 3 and Appendix A). Even more diseases have been addressed in studies related to Nrf2 since 2011, namely cardiovascular disease, obesity, insulin resistance and diabetes, stroke, and various types of cancer (most notably, breast, hepatocellular, and lung carcinoma) (Figure 4 and Figure 5, Appendix A).

### 3.3. Evolution of Nrf2-Related Research Areas and its Countries of Origin

The main research areas of Nrf2 papers are “biochemistry/molecular biology”, “pharmacology/pharmacy”, “cell biology” and “toxicology” as they are always in the top five research areas over the years (Figure 6A–D). Even though these areas were dominant, especially from 1990 to 2005 and from 2006 to 2010, as the number of publications increased, the Nrf2 research areas became more diverse, including papers in the fields of “endocrinology/metabolism”, “neuroscience/neurology”, “oncology”, “research experimental medicine”, “food science technology” and “immunology”. This shows a trend towards more translational-clinical research on Nrf2 as major basic science data on the Nrf2 pathway accumulated. Restricting the “Web of Science” to Nrf2 papers from 1990 to 2019 containing the phrase “clinical trial” yielded 72 records, 47 of which (~2/3) were published from 2015 to today (July 2019). This indicates that the interest in clinical applications of Nrf2 has recently expanded and culminated in the realization of clinical trials with Nrf2 pathway activators, including e.g., the use of sulforaphane-rich broccoli sprout extracts for detoxication of airborne pollutants [23,42] or for diabetes [25], bardoxolone methyl (CDDO-Me) for chronic kidney disease [27] and dimethyl-fumarate (BG-12) for multiple sclerosis [43].

Initially (between 1990 and 2005), papers about Nrf2 originated mainly in the USA (~60%) and Japan (~30%) (Appendix A). Gradually, awareness of Nrf2 increased in the scientific community, knowledge on the pathway accumulated, and work on Nrf2 also became more feasible, thanks to the development of animal models. Thus, with time, the contribution of additional countries to Nrf2 research became more substantial in terms of the volume of productivity. Specifically, between 2006 and 2010, the USA participated in ~48% of published Nrf2 papers and Japan in ~19%. Countries like South Korea (~12%), China (~9%), UK (~6%) and Germany (~5%) started to produce considerable amount of Nrf2-related research (Appendix A). From 2011 to 2015, China further increased its participation in the Nrf2 research output (~26% of total papers), with the USA still being the major contributor with ~33%, and Japan and South Korea producing ~10% each. European countries and other areas increased their share (Appendix A). In the latest time period (2016–2019) output from China increased to a great degree now participating in ~45% of Nrf2 papers, while the USA contributes to 21%. South Korea and Japan follow with 9% and 6.5%, respectively (Appendix A). Germany, India, UK, and Italy come next with ~4% each, followed by Brazil and Spain with 2.7% each. Hence, Nrf2 research is now being performed worldwide and it is noteworthy that a large volume of papers now comes from China. As an index of relative impact, Figure 7 depicts citations per Nrf2-related paper in each country with at least 20 published papers for the whole period of analysis (1990–July 2019). The leading countries in this aspect are Japan, the UK, and the USA, followed by Switzerland, Norway and Belgium. Appendix A includes the actual citations per document for each country (averaged citations) next to the total number of publications.

To obtain a better understanding of the contributions of each country to the Nrf2 research field and the collaborations between countries, data for the whole period (1990–2019) were analyzed with a focus on international collaborations, the times each country cites the work of another, and the number of citations these papers have received. As depicted in Figure 8A, China, the USA, South Korea and Japan have the highest numbers of Nrf2-related publications (bigger bubbles), but the USA and Japan have attracted the most citations per document (red bubbles). The UK (GBR), Norway and Switzerland also have high averaged citation counts, followed by Belgium, the Netherlands, Canada, and Germany. The USA and the UK have a high number of co-authored papers (small distance between the respective bubbles). In Figure 8B, the smaller the distance between two bubbles, the higher the frequency with which each country’s Nrf2-related publications cite those of the other. For example, the USA, the UK, Japan, Switzerland, and Germany are in close proximity, meaning they cite each other very often. The color of the bubble in Figure 8B indicates the average year of Nrf2-related publications of the respective country and shows that some countries have only recently entered the Nrf2 field (e.g., red color for Brazil, India, and Egypt).

### 3.4. Natural Compounds in Nrf2-Related Research

Figure 9 illustrates natural products featured in Nrf2-related research published from 1990 to 2019. As expected from the previous analyses, sulforaphane has the highest occurrence frequency (337 papers, Appendix A). Curcumin and resveratrol show also high occurrence frequencies, as do the more general categories of isothiocyanates and flavonoids (Figure 9). Most of these compounds are Nrf2 pathway activators. Of note, brusatol, an inhibitor of the Nrf2 pathway also appears in the list (with 12 occurrences), apparently reflecting the more recent interest in research into Nrf2 inhibitors as means to treat cancer or to increase its sensitivity to chemotherapy [44].

## 4. Discussion

### 4.1. Nrf2-Related Research: Expanding and Evolving from the Bench to the Bedside

This bibliometric analysis of Nrf2-related literature to date revealed some interesting and useful facts. Firstly, the field of Nrf2-related research is continuously expanding, as reflected in both the accelerating pace of new Nrf2-related publications (currently close to 2000 per year) and the increasing variety of research areas involved. Specifically, Nrf2 papers are no longer limited to basic research categories (e.g., biochemistry/molecular biology, cell biology, pharmacology, and toxicology), but they also encompass more clinical areas (e.g., oncology, neurology, endocrinology and metabolism, etc.) (Figure 6). This trend is also reflected in the fact that gradually, since 2006, more terms appear related to clinical topics like chemoprevention, respiratory diseases like chronic obstructive pulmonary disease, neurodegenerative diseases like Alzheimer’s and Parkinson’s, and metabolic disorders like insulin resistance and diabetes (Figure 2, Figure 3, Figure 4 and Figure 5). This trend towards translational-clinical research on Nrf2 is also clearly evident in the term maps of the most recent Nrf2 papers (2016–2019) (Figure 5), which highlight groups of disease-related terms that co-appear in publications. For example, one such cluster of terms is related to neurological diseases (“Parkinson’s disease”, “Alzheimer’s disease”, “neurodegeneration”, etc.), for which activation of the Nrf2 pathway is considered a highly promising target [16]. Most importantly, the Nrf2 pathway activator BG-12 has been approved by the USA Food and Drug Administration and by the European Medicines Agency for adults with remitting-relapsing multiple sclerosis [43]. Moreover, the interest in Nrf2 in clinical cancer research is also highlighted in Figure 5 by the clustering of terms such as “cancer”, “breast cancer”, “hepatocellular carcinoma” and “lung cancer”. Pharmacological Nrf2 pathway activation has been considered a means of chemoprevention against the development of cancer [22,45]. However, during the process of carcinogenesis, somatic mutations (e.g., in the gene encoding Keap1) and other epigenetic events constitutively activate the Nrf2 pathway and potentially confer resistance to chemotherapeutic drugs [37,46]. Other closely related terms that tend to cluster together in Figure 5 are “obesity”, “insulin resistance”, “diabetes” and “hyperglycemia”. These indicate research on Nrf2 in the field of the metabolic syndrome and diabetes. A series of preclinical research studies have suggested the activation of the Nrf2 pathway as a means to target obesity and insulin resistance [47,48,49]. A clinical trial showed that treatment of patients with uncontrolled type 2 diabetes with sulforaphane-rich broccoli sprout extract reduces the levels of fasting glucose and glycated hemoglobin [25]. Based on studies in a variety of experimental animal models, Nrf2 has also been considered a target in chronic kidney disease [34]. The clustering of the terms “kidney”, “nephrotoxicity”, and “acute kidney injury” in Figure 5 is indicative of this. A clinical trial with the synthetic Nrf2 pathway activator CDDO-Me in patients with type 2 diabetes and chronic kidney disease showed amelioration in the glomerular filtration rate [50]. However, a second trial was terminated early due to an increase in cardiovascular events in the CDDO-Me group [27] for reasons that warrant further investigation [51]. CDDO-Me is currently being tested in patients with Alport syndrome (clinical trial: NCT03019185). These are just some indicative examples of translational-clinical areas where Nrf2-related research has entered. The temporal trend of terms encountered in Nrf2-related research also underlines the steady and gradual inclusion of clinical research. For example, the terms “mouse” and “rat” appear from the early days of Nrf2-related studies (Figure 2), reflecting basic research using these models. Lately, more terms appear that are related to clinical conditions, as discussed above (e.g., neurodegeneration, diabetes, etc.). For example, the clustering of “dimethyl fumarate” with “microglia”, “neuroinflammation” and “brain” (Figure 5, Appendix A) is consistent with the recently approved clinical use of BG-12 for multiple sclerosis [26].

### 4.2. Nrf2-Related Studies Are Published in Both Basic and Clinical Journals

The list of journals that have published at least 20 papers on Nrf2 from 1990 to 2019 (Appendix A) indicates that the number and the thematic range of journals that publish Nrf2-related research are also expanding. For example, the top of the list includes journals like “*Free Radical Biology and Medicine*”, “*Journal of Biological Chemistry*” and “*Oxidative Medicine and Cellular Longevity*” that mainly publish basic studies or preclinical studies with mouse or rat models. In the top 10 are also two major open access journals that publish research from any discipline (“*PLoS One*” and “*Scientific Reports*”) and “*Biochemical and Biophysical Research Communications*” that publishes short but innovative research reports. Pharmacology-toxicology journals also publish a substantial proportion of Nrf2-related research, reflecting not only the interest in Nrf2 as a drug target but also the defensive roles of Nrf2 against toxic insults. Examples of such journals in the top 10 include “*Food and Chemical Toxicology*”, “*Toxicology and Applied Pharmacology*” and “*Biomedicine and Pharmacotherapy*”. The list also includes journals dedicated to specific medical fields (most notably, neurology, oncology, endocrinology, and nephrology), reflecting the attention that Nrf2 has started to attract from experimental medicine and from the various clinical specialties (Appendix A). This is another indication of the broadening scope of Nrf2-related studies, which now cover both the bench and the bedside.

### 4.3. Natural Compound Modulators of Nrf2 with Clinical Implications

Natural compounds that activate the Nrf2 pathway were among the frequently occurring terms, including the general category of isothiocyanates, the isothiocyanate sulforaphane, curcumin, resveratrol, etc. (Figure 4, Appendix A). Therefore, a dedicated analysis was performed to formally address which natural products are most frequently encountered in Nrf2-related research (Figure 9). The results are consistent with the fact that the most commonly used approach in disease prevention studies centered on the Nrf2 pathway remains the activation of Nrf2 by sulforaphane as a strategy for cancer chemoprevention [45]. At the same time, the results indicate that the research community has started to become interested not only in natural compounds that activate the Nrf2 pathway [52], but also in Nrf2 inhibitors, such as brusatol, which is also a naturally occurring substance. This likely reflects the fact that, in some pathologies, such as lung cancer, inhibition of the Nrf2 pathway could be beneficial, as it could reduce the aggressiveness of the tumor and/or increase its responsiveness to standard chemotherapy [53,54,55].

### 4.4. A Changing Global Landscape of Nrf2-Related Research

Another interesting aspect of the bibliometric analysis is the visualization of global trends in the output of Nrf2-related research papers, as illustrated in Figure 7. The main observation arising from this analysis was the gradual shift of the majority of Nrf2-related research output, in terms of volume, from the USA and Japan to China. It should be noted that this does not mean that the USA and Japan produce less Nrf2-related research in absolute values as compared to their past outputs. Rather, it is their relative contribution to Nrf2-related research papers as a percentage of the total global output that has decreased due to China’s increased research output. European countries and other Asian countries also make substantial contributions to the total Nrf2-related research output.

### 4.5. Limitations and Strengths of the Bibliometrics Approach

One limitation of this type of bibliographic analysis is that this approach reflects only the quantity of research papers, the frequency of occurrence of specific terms, the research output of various countries and the citations their papers receive. It thus does not assess the quality of the respective publications, neither does it correct for the factors such as a country’s population size, total research output, the gross domestic product, resources invested in research, etc. These considerations should be taken into account when interpreting or communicating the countries data emanating from the present bibliographic analysis. As a relative measure of quality, impact or usefulness of the published papers, the averaged citation count was calculated for each country (Figure 7, Figure 8A). This is the average number of citations received by the papers published by each country. The USA, Japan, the UK, Norway, and Switzerland are leaders in this metric (Figure 8A, red bubbles). Other European countries like Belgium, the Netherlands, Germany, Ireland, France, Denmark, and Sweden follow along with Canada. China and South Korea have not yet attracted a comparable number of citations. However, the average year of Nrf2-related publications is more recent in China (~2014) compared to the USA (~2012) or Japan (~2011) (Figure 8B), indicating that researchers based in China entered the Nrf2 field later.

Nevertheless, the advantage of this type of analysis is that it is unbiased. It can illustrate where the Nrf2 field has been heading and it can also highlight areas where Nrf2 has not yet been thoroughly investigated.

## 5. Conclusions

In conclusion, the present analysis revealed that Nrf2 is in the spotlight of research worldwide and it is expected that we will continue to see an increasing research output in the near future. Nrf2-focused research started from basic science but has expanded to encompass multiple clinical perspectives. A particular clinical focus is on natural compounds that modify Nrf2 activity, whose pharmacological applications in health and disease are being actively explored.

## Figures and Tables

**Figure 1 antioxidants-08-00353-f001:**
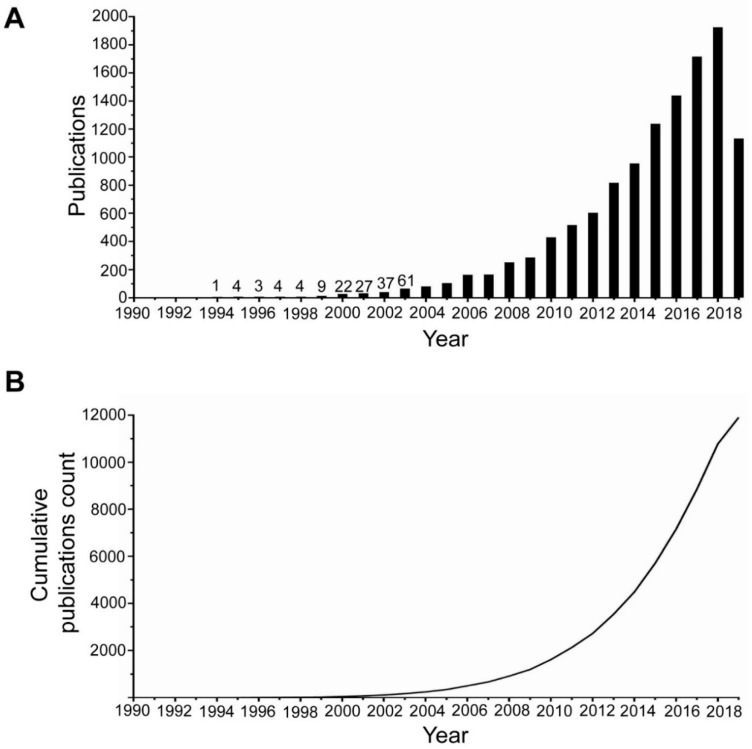
(**A**) Number of publications about Nrf2 per year. The numbers on top of the first bars indicate the exact number of publications for the respective year. Data up to July 2019. (**B**) Cumulative number of publications about Nrf2 per year. Data up to July 2019.

**Figure 2 antioxidants-08-00353-f002:**
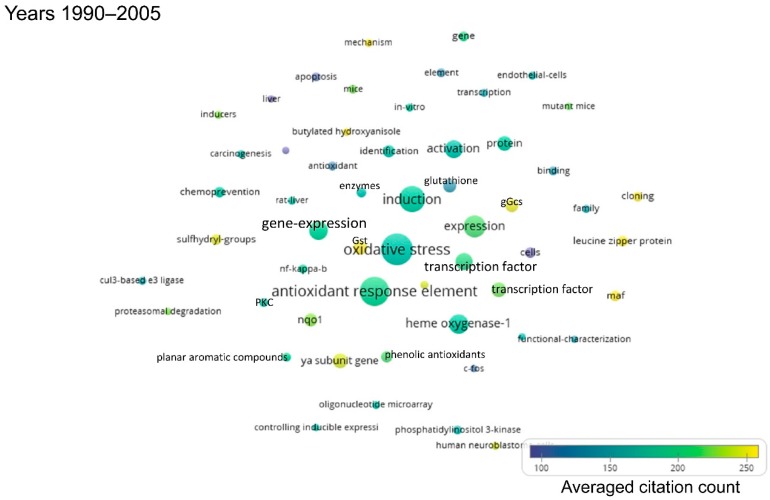
Term map for years 1990–2005. Bubble map showing the visualization of 55 terms that appeared in at least 10 publications published between 1990 and 2005. A bubble’s size indicates the occurrence frequency of the words (the bigger the bubble, the higher the frequency). Multiple occurrences in a single publication count as one. A bubble’s color indicates the averaged citations count received by publications containing the indicated word (i.e., the average number of citations received by the documents in which a term occurs). The proximity of two bubbles indicates the frequency of co-occurrence between the two respective words (the closer the proximity, the higher the frequency). All the terms visualized are listed in Appendix A, along with their respective occurrence frequencies and averaged citations. Gst: Glutathione s-transferase, gGcs: γ-glutamylcysteine synthetase, PKC: Protein kinase C.

**Figure 3 antioxidants-08-00353-f003:**
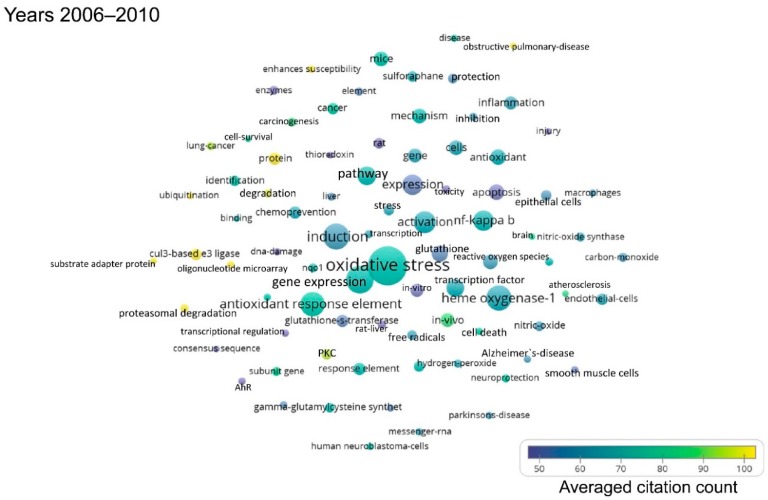
Term map for years 2006–2010. Bubble map showing the visualization of 83 terms that appeared in at least 20 publications published between 2006 and 2010. For explanations on bubble size, color and proximity, see the legend of Figure 2. All the terms visualized are listed in Appendix A, along with their respective occurrence frequencies and averaged citations. AhR: Aryl hydrocarbon receptor, PKC: Protein kinase C.

**Figure 4 antioxidants-08-00353-f004:**
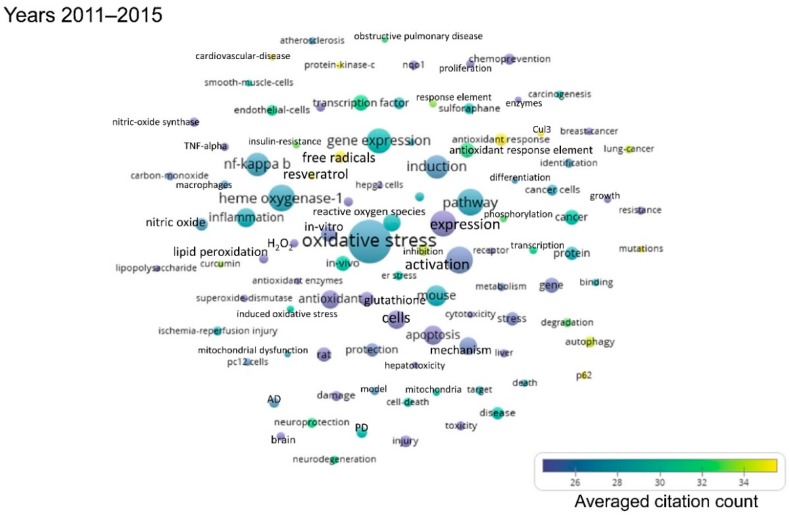
Term map for years 2011–2015. Bubble map showing the visualization of 100 terms that appeared in at least 50 publications published between 2011 and 2015. For explanations on bubble size, color, and proximity, see the legend of Figure 2. All the terms visualized are listed in Appendix A, along with their respective occurrence frequencies and averaged citations. AD: Alzheimer’s disease, Cul3: Cul3-based E3 ligase, H_2_O_2_: Hydrogen peroxide, PD: Parkinson’s disease.

**Figure 5 antioxidants-08-00353-f005:**
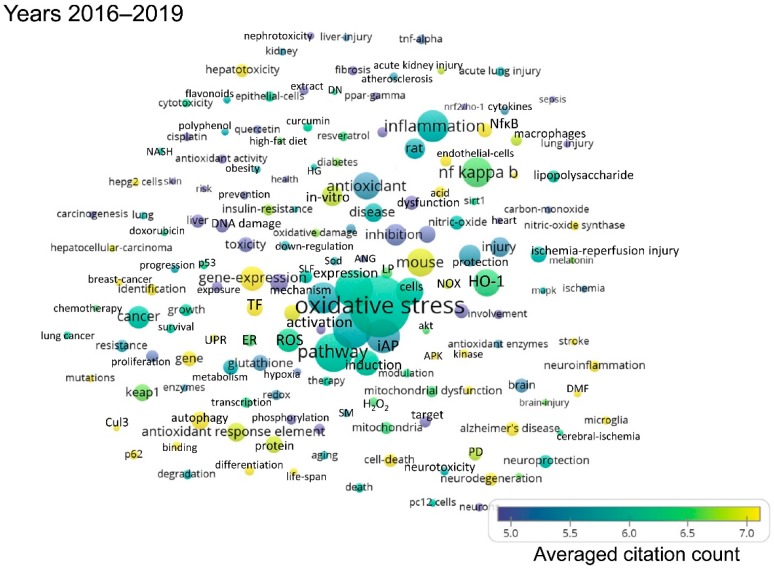
Term map for years 2016–2019. Bubble map showing the visualization of 140 terms that appeared in at least 50 publications published between 2016 and July 2019. For explanations on bubble size, color and proximity, see the legend of Figure 2. All the terms visualized are listed in Appendix A, along with their respective occurrence frequencies and averaged citations. ANG: Angiogenesis, APK: Activated protein kinase, Cul3: Cul3-based E3 ligase, DMF: Dimethyl fumarate, DN: Diabetic nephropathy, H_2_O_2_: Hydrogen peroxide, HG: Hyperglycemia, HO-1: Heme oxygenase-1, iAP: Induced apoptosis, LP: Lipid peroxidation, NASH: Non-alcoholic steatohepatitis, NOX: NADPH oxidase, PD: Parkinson’s disease, ROS: Reactive oxygen species, SLF: Sulforaphane, Sod: Superoxide-dismutase, TF: Transcription factor.

**Figure 6 antioxidants-08-00353-f006:**
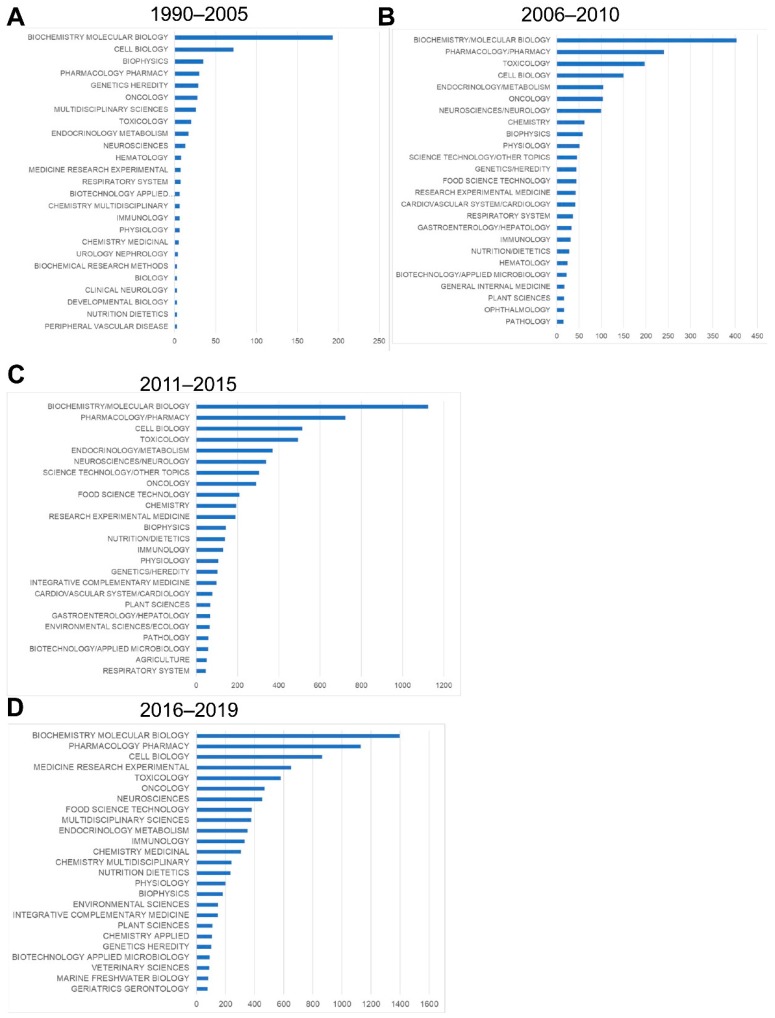
Top 25 research areas of Nrf2-related studies per indicated time period. (**A**) 1990–2005. (**B**) 2006–2010. (**C**) 2011–2015. (**D**) 2016–2019. Data up to July 2019. Each bar indicates the number of Nrf2-related papers falling into the respective research area. Each paper can belong to more than one research areas.

**Figure 7 antioxidants-08-00353-f007:**
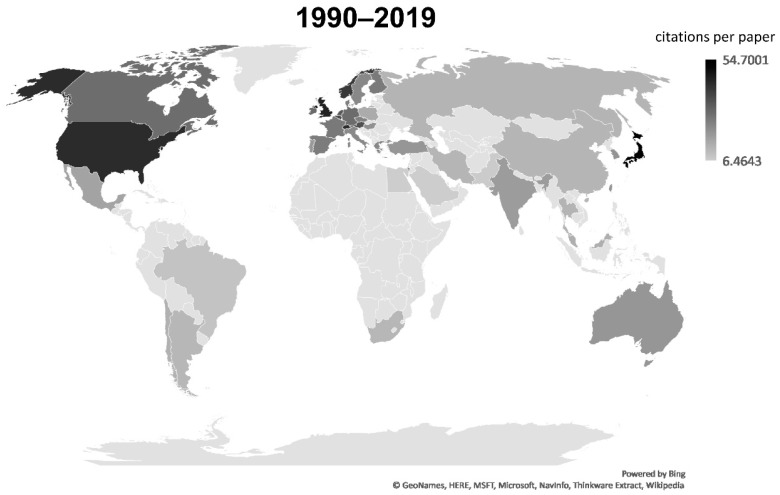
World map depicting the citations per Nrf2-related paper (averaged citations) published from each country from 1990 to July 2019. Each paper may have authors from more than one country (international collaboration). The higher the color density (darker grey to black) of a country, the larger the number of citations per Nrf2 paper. Note that “England”, “Scotland”, “Wales” and “Northern Ireland” records from the “Web of Science” were combined into “UK”. Only countries with at least 20 publications are depicted.

**Figure 8 antioxidants-08-00353-f008:**
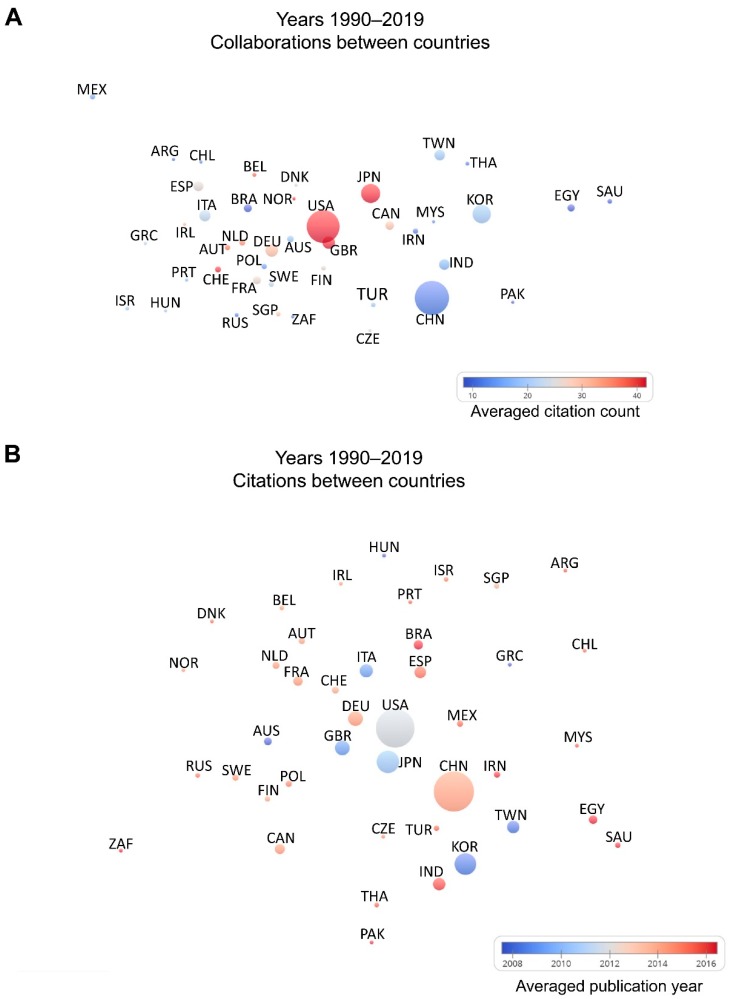
Country term maps for years 1990–2019. (**A**) Bubble map showing collaborations between countries on Nrf2-related research. Countries with at least 20 publications are included. The smaller the distance between two bubbles, the higher the frequency with which these two countries appear in the same paper in the author list. A larger bubble indicates a higher number of publications. A bubble’s color indicates the average number of citations received by the papers published by a country. (**B**) Bubble map showing the frequency with countries cite each other in their Nrf2-related papers. Countries with at least 20 publications are included. The smaller the distance between two bubbles, the higher the frequency with which each country cites the other in Nrf2-related papers. A larger bubble indicates a higher number of publications. A bubble’s color indicates the average publication year of the Nrf2-related papers published by a country. Note that “England”, “Scotland”, “Wales” and “Northern Ireland” records from “Web of Science” were combined into “UK”. Countries are presented with their alpha-3 code based on the ISO 3166 international standard.

**Figure 9 antioxidants-08-00353-f009:**
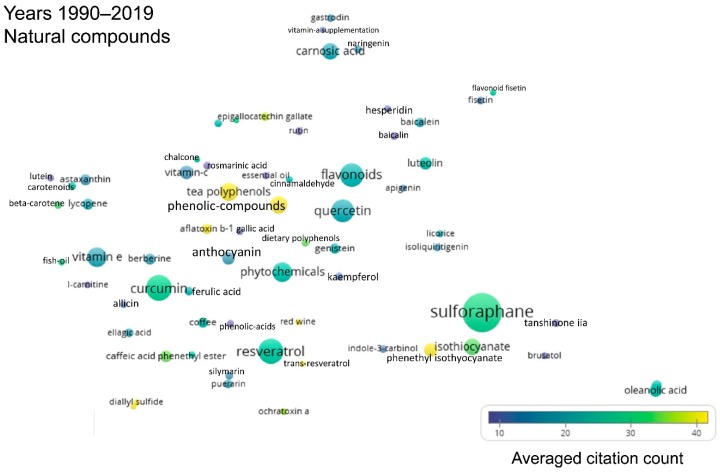
Term map for natural products related to Nrf2 for years 1990–2019. Bubble map showing the visualization of 66 natural product-related terms that appeared in at least 10 papers published between 1990 and July 2019. For explanations on bubble size, color and proximity, see the legend of Figure 2. All the terms visualized are listed in Appendix A, along with their respective occurrence frequencies and averaged citations (average number of citations received by the documents in which a term occurs).

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
