# Peer review of "A Bibliometric Review of the Keap1/Nrf2 Pathway and its Related Antioxidant Compounds"

_antioxidants, 2019, doi:10.3390/antiox8090353_

Round 1
Reviewer 1 Report
The authors present a bibliometric review of the Nrf2 pathway and its related antioxidant compounds. To achieve this, the “Web of Science” database was extensively evaluated using common Nrf2-related terms and research volume output, amongst other determining factors. The main findings are that Nrf2-related research is evolving from basic to translational-clinical research, and that the relative contribution of countries to the Nrf2 field is changing. Whilst interesting for those in the field, this review lacks content that could be useful to readers, particularly newcomers to the field. As a result, there are several areas for improvement, as detailed below.
Major points
The authors should provide a more detailed analysis of the specific diseases/pathologies associated with Nrf2 research over the years. This would highlight popular and niche areas of translational potential, and hence inform readers’ decisions on research direction.
It would be appropriate to include a more detailed analysis of the evolution of Nrf2 research from basic biology to translation/clinical over theyears. This could be in the form of a bibliometric review of the volume of work published in pre-clinical (in vitro/in vivo) and clinical (experimental medicine and clinical trials) settings.
Minor points
Page 2, line 52. Remove first instance of ‘research’.
Page 2, line 82. Replace ‘they’ with ‘that’.
Page 3, figure 1. Consider excluding 2019 data from the graph as it does not represent a full calendar year.
Page 4, line 113. The authors should explain why glutathione and glutathione-related enzymes are included as common, Nrf2-related terms.
Page 4, figure 2, line 116. Replace ‘at least in 10 publications’ with ‘in at least 10 publications’. Similar for figures 3 (page 5, line 135), 4 (page 6, line 148) and 5 (page 6, line 153).
Page 9, line 192. Replace ‘rectangles’ with ‘bubbles’.
Page 9, figure 7. Correct figure legend (currently labelled as figure 1) and improve readability of maps (increase size, use colour).
Use of English could be improved throughout the manuscript.
Author Response
Reviewer 1
The authors present a bibliometric review of the Nrf2 pathway and its related antioxidant compounds. To achieve this, the “Web of Science” database was extensively evaluated using common Nrf2-related terms and research volume output, amongst other determining factors. The main findings are that Nrf2-related research is evolving from basic to translational-clinical research, and that the relative contribution of countries to the Nrf2 field is changing. Whilst interesting for those in the field, this review lacks content that could be useful to readers, particularly newcomers to the field. As a result, there are several areas for improvement, as detailed below.
Response: We thank the Reviewer for carefully assessing our manuscript and for his/her insightful comments. We have responded to these comments below.
Major points
The authors should provide a more detailed analysis of the specific diseases/pathologies associated with Nrf2 research over the years. This would highlight popular and niche areas of translational potential, and hence inform readers’ decisions on research direction.Response: We appreciate the Reviewer’s comment. We were pleased to include dedicated sections (4.1 and 4.2) in the Discussion to explain in more details some of the most important clinical/translational research works on Nrf2, as they emerge from the bibliometric analysis.
It would be appropriate to include a more detailed analysis of the evolution of Nrf2 research from basic biology to translation/clinical over the years. This could be in the form of a bibliometric review of the volume of work published in pre-clinical (in vitro/in vivo) and clinical (experimental medicine and clinical trials) settings.Response: We understand the Reviewer’s concern but we are unable to perform a bibliometric analysis of preclinical vs. clinical work, because there are no specific tags in the abstract of each paper to allow the bibliometry software (Web of Science or VOS viewer) to clearly identify each work as preclinical or experimental/clinical medicine. What we were able to do, was to perform a new search of the Web Of Science about Nrf2, including the phrase “clinical trial”. We found 72 records, with 2/3 of them published in the last four years, which is quite striking. We describe this in the revised Results, section 3.3.
Minor points
Page 2, line 52. Remove first instance of ‘research’.Response: Done, thank you!
Page 2, line 82. Replace ‘they’ with ‘that’.Response: Done, thank you!
Page 3, figure 1. Consider excluding 2019 data from the graph as it does not represent a full calendar year.Response: We appreciate the Reviewer’s feedback on this. However, as in all the following figures we include 2019 data (up to July 2019), we preferred to keep this information in Figure 1 as well, clearly stating in the legend that the data are up to July 2019.
Page 4, line 113. The authors should explain why glutathione and glutathione-related enzymes are included as common, Nrf2-related terms.Response: The Reviewer is right that it is not clear to a reader not directly working on the Nrf2 field why the term “glutathione” appears so often in Nrf2-related articles. To address this, we have expanded the Introduction, mentioning a few examples of Nrf2 target genes, including the ones related to glutathione, along with the relevant references as follows: “…sulforaphane) [1-3]. Targets of Nrf2 include glutathione synthesis/recycling genes (e.g. γ-glutamylcysteine synthetase currently known as glutamate-cysteine ligase) [4,5], genes involved in detoxication/conjugation reactions (e.g. glutathione s-transferase) [6], antioxidative genes (e.g. NAD(P)H quinone dehydrogenase 1) [7] and proteasome units [8].”
We also modified the text on page 4 as follows: “this is evident from the relevant terms “oxidative stress”, “antioxidant response element”, “antioxidant”, “glutathione” and cytoprotective enzymes such as “glutathione s-transferase”, “γ-glutamylcysteine synthetase”, “NAD(P)H quinone dehydrogenase 1 (Nqo1)”.
Page 4, figure 2, line 116. Replace ‘at least in 10 publications’ with ‘in at least 10 publications’. Similar for figures 3 (page 5, line 135), 4 (page 6, line 148) and 5 (page 6, line 153).Response: Done, thank you!
Page 9, line 192. Replace ‘rectangles’ with ‘bubbles’.Response: Done, thank you!
Page 9, figure 7. Correct figure legend (currently labelled as figure 1) and improve readability of maps (increase size, use colour).Response: We appreciate the Reviewer’s comments. We have replaced the multipanel Figure 7 with a new, bigger world map that shows the averaged citation count for each country. The former figure 7 has moved in the supplemental data as 4 independent figures (Fig. S1-S4); these are now more readable. We have also included an extra panel (Panel B) for each new figure, showing the world map with the averaged citation counts.
Use of English could be improved throughout the manuscript.Response: We worked on the whole manuscript again, correcting and simplifying the language.
Reviewer 2 Report
This manuscript reviews some aspects of the current Nrf2 field. The authors present information about the increase in Nrf2 publications since 1994. There is some beneficial information in the review, particularly the compounds most often used in Nrf2 studies. However, there are also some aspects of the review that should be changed as described below:
The information about countries from which Nrf2 publications originate should be removed. As is, this does not add much to our understanding of the Nrf2 field and the authors were not able to account for quality of publications.
The term maps and natural compound maps would be much easier to assess and glean information from if they were in tables. As is, the figures are difficult to interpret and this information would be important for interested researchers.
The statement “Nrf2 defines a globally active research field” is used twice and should be revised. The word “defines” should be changed to “is the subject of” or something similar.
Author Response
Reviewer 2
This manuscript reviews some aspects of the current Nrf2 field. The authors present information about the increase in Nrf2 publications since 1994. There is some beneficial information in the review, particularly the compounds most often used in Nrf2 studies. However, there are also some aspects of the review that should be changed as described below:
Response: We express our gratitude to Reviewer 2 for carefully evaluating our manuscript and for his/her positive evaluation. We respond point by point to his/her comments below.
The information about countries from which Nrf2 publications originate should be removed. As is, this does not add much to our understanding of the Nrf2 field and the authors were not able to account for quality of publications.Response: It is true that the country of origin for Nrf2 articles was not the main focus of the present bibliographic review. However, we wanted to include this information to show mainly that research on Nrf2 is now being performed worldwide. It was not our purpose to make generalizations about the quality of the scientific research performed in each country, neither did we want to confuse quantity with quality. We clarified these issues in the revised manuscript. We used the “averaged citation count” as a relative index of impact; this metric is the average number of citations per article for each country. We replaced figure 7 with a world map that shows the averaged citation count and we have moved the 4 panels of former figure 7 in the supplemental data as 4 independent figures; as a result, the figures are now also more readable.
The term maps and natural compound maps would be much easier to assess and glean information from if they were in tables. As is, the figures are difficult to interpret and this information would be important for interested researchers.Response: We agree with the Reviewer that it would be useful for a reader who is interested in gleaning detailed information to have access to tables with all the terms. As these tables are too lengthy for the main text and we have included them as Supplementary Tables.
The statement “Nrf2 defines a globally active research field” is used twice and should be revised. The word “defines” should be changed to “is the subject of” or something similar.Response: We thank the Reviewer for spotting this redundancy. We have changed the wording of these phrases and eliminated the repetition.
Reviewer 3 Report
This manuscript provides a comprehensive review on the research on Nrf2, a key regulator of antioxidant pathways.
(1) This reviewer found Discussion somewhat lengthy and redundant, especially the discussion on China’s increased output (L272–290). The surge in China’s output both in absolute number of publications and in relative percentage in recent years is clear in Figure 7, and described in the Results.
(2) The ‘quality’ and ‘impact’ of research (L303) can be better judged if the authors count citations per paper per year, not cumulative citations per paper. Also, if the authors restrict their citation counting to more recent publications (20016–2019), a better picture of relative impact of research done in different countries will emerge.
(3) The detailed description of literature survey methods will be useful to those who are not familiar with literature analysis.
(4) In Figures 2–5 and 9, there are several unannotated bubbles, likely due to space limitation. More abbreviations can be used, as in Fig 5.
Minor typographical suggestions.
L49: [21].
L92 and elsewhere: 1994–2019. An ‘en dash’, not a hyphen, for a range.
L122: gGcs;
L139: chemoprevention,
L173: easy to work on
L177: Table S6).
L187: received, we
L190: counts
L200: Figure 7
L255: performed
Author Response
Reviewer 3
This manuscript provides a comprehensive review on the research on Nrf2, a key regulator of antioxidant pathways.
Response: We thank the Reviewer for carefully evaluating our manuscript and for the useful comments. We respond to them below.
(1) This reviewer found Discussion somewhat lengthy and redundant, especially the discussion on China’s increased output (L272–290). The surge in China’s output both in absolute number of publications and in relative percentage in recent years is clear in Figure 7, and described in the Results.
Response: We agree with the Reviewer that mentioning too many numbers is not necessary as the readers can resort to the relevant supplementary tables for details. We have significantly edited this part of the Discussion by shortening it about 50% and just giving some highlights on the countries contribution to Nrf2-related research recently (2016-2019), Cf. page 14 section 4.3.
(2) The ‘quality’ and ‘impact’ of research (L303) can be better judged if the authors count citations per paper per year, not cumulative citations per paper. Also, if the authors restrict their citation counting to more recent publications (20016–2019), a better picture of relative impact of research done in different countries will emerge.
Response: We agree with the Reviewer that it is hard to assess the quality of the research papers published and we did not intend to do so. To assess the impact of papers per country, we already had the metric of averaged citation count, meaning citations per article for each country (Fig. 8A, as indicated by the bubble color). The VOSViewer tool does not offer the metric of citations per paper per year for each country. We understand that the information on citations is very useful. This is the reason why we have replaced figure 7 (depicting the absolute number of publications for each country per time period) with a new figure that shows a map with the averaged citation count. The former figure 7 has now been moved to the supplementary section and has been divided in 4 independent figures (one for each time period; 1990-2005, 2006-2010, 2011-2015, 2016-2019) with 2 panels each. In each of these supplemental figures, Panel A shows the world map with the absolute number of publications per country; panel B shows the averaged citation count per country for every given period. In this way, the maps are larger and more legible. Moreover, we have enriched tables S5, S6, S7, S8, that contained the absolute publications number per country and the % contribution, with the averaged citation count (citations per document). As can be seen from these data, just calculating the citations per document (adjusted or not by time) cannot be an absolute marker of impact. For example, Ecuador, with 2 publications between 2016-2019 has an average of 46.5 citations per publication. The USA, with 1279 publications has 8.09 citations per publication; China, with 2815 publications has 5.35 citations per publication; and Japan, with 391 publications, has 6.64 citations per publication. In the new table S9, we include the information about the total number of publications and citations per publication for the whole period 1990-2019. In the revised manuscript, we have explained more clearly what is being assessed (quantity of productivity, and a metric of its impact). This will avoid confusion or misunderstandings about the different countries.
(3) The detailed description of literature survey methods will be useful to those who are not familiar with literature analysis.
Response: We agree, and we included more details in Materials and Methods, section 2.1.
(4) In Figures 2–5 and 9, there are several unannotated bubbles, likely due to space limitation. More abbreviations can be used, as in Fig 5.
Response: We agree on this. The software automatically removes some annotations to make space. We have now manually added the annotations in all these figures. On top of this, in Figure 9 showing the natural compounds we removed panel B (it was the same as panel A, with the exception that it indicated the year of publication; this avoids the redundancy and allows to enlarge panel A). We preferred to keep panel A that shows the averaged citation number per compound.
Minor typographical suggestions.
L49: [21]. Response: Fixed, thank you! Please note the reference number changed due to inclusion of more references.
L92 and elsewhere: 1994–2019. An ‘en dash’, not a hyphen, for a range. Response: Fixed, thank you!
L122: gGcs. Response: Fixed in images and figure legends, thank you!
L139: chemoprevention. Response: Fixed, thank you!
L173: easy to work on. Response: Fixed, thank you!
L177: Table S6). Response: Fixed, thank you!
L187: received, we. Response: Fixed, thank you!
L190: counts. Response: Fixed, thank you!
L200: Figure 7. Response: Fixed, thank you!
L255: performed. Response: Fixed, thank you!
Round 2
Reviewer 1 Report
The authors have addressed the issues raised in my initial review.
Author Response
We thank Reviewer 1 for carefully evaluating our manuscript.
Reviewer 2 Report
My concerns were addressed.
Author Response
We thank Reviewer 2 for his/her thorough evaluation of our manuscript.
Reviewer 3 Report
All numbers in Tables should have the same number of significant figures. I suggest all data are reported as numbers with one decimal digit. Not 234.5678, but 234.6. In this way, numbers in a column will align for easy comparison.
Author Response
We thank Reviewer 3 for the careful evaluation of our manuscript and for the suggestion about the tables. We have fixed this issue in the revised tables.